# Remotely Sensed Land Surface Temperature-Based Water Stress Index for Wetland Habitats

**Wojciech Ciężkowski [1],\*, Sylwia Szporak-Wasilewska [2], Małgorzata Kleniewska [1], Jacek Jóźwiak[2], Tomasz Gnatowski [3], Piotr Dąbrowski [3], Maciej Góraj [†], Jan Szatyłowicz [3], Stefan Ignar [1] and Jarosław Chormański [1]**

[1]  Department of Remote Sensing and Environmental Assessment, Institute of Environmental Engineering, Warsaw University of Life Science – SGGW, Nowoursynowska 166, 02-787 Warsaw, Poland; malgorzata_kleniewska@sggw.pl (M.K.); s.ignar@levis.sggw.pl (S.I.); j.chormanski@levis.sggw.pl (J.C.)

[2]  Water Center Laboratory, Warsaw University of Life Sciences - SGGW, Nowoursynowska 166, 02-787 Warsaw, Poland; s-szporak@levis.sggw.pl (S.S.), j.jozwiak@levis.sggw.pl (J.J.)

[3]  Department of Environmental Improvement, Institute of Environmental Engineering, Warsaw University of Life Sciences - SGGW, Nowoursynowska 166, 02-787 Warsaw, Poland; tomasz_gnatowski@sggw.pl (T.G.); piotr_dabrowski@sggw.pl (P.D.), jan_szatyłowicz@sggw.pl (J.S.)

[†]  Deceased 11 January 2019.

[\*]  Correspondence: w.ciezkowski@levis.sggw.pl

**Abstract:** Despite covering only 2%–6% of land, wetland ecosystems play an important role at the local and global scale. They provide various ecosystem services (carbon dioxide sequestration, pollution removal, water retention, climate regulation, etc.) as long as they are in good condition. By definition, wetlands are rich in water ecosystems. However, ongoing climate change with an ambiguous balance of rain in a temperate climate zone leads to drought conditions. Such periods interfere with the natural processes occurring on wetlands and restrain the normal functioning of wetland ecosystems. Persisting unfavorable water conditions lead to irreversible changes in wetland habitats. Hence, the monitoring of habitat changes caused by an insufficient amount of water (plant water stress) is necessary. Unfortunately, due to the specific conditions of wetlands, monitoring them by both traditional and remote sensing techniques is challenging, and research on wetland water stress has been insufficient. This paper describes the adaptation of the thermal water stress index, also known as the crop water stress index (CWSI), for wetlands. This index is calculated based on land surface temperature and meteorological parameters (temperature and vapor pressure deficit – VPD). In this study, an unmanned aerial system (UAS) was used to measure land surface temperature. Performance of the CWSI was confirmed by the high correlation with field measurements of a fraction of absorbed photosynthetically active radiation (R = –0.70) and soil moisture (R = –0.62). Comparison of the crop water stress index with meteorological drought indices showed that the first phase of drought (meteorological drought) cannot be detected with this index. This study confirms the potential of using the CWSI as a water stress indicator in wetland ecosystems.

**Keywords:** drought monitoring; CWSI; LST; thermal infrared; UAS; Biebrza River Valley; Janów Forest Landscape Park

## 1. Introduction

Wetland ecosystems cover only around 4% (depending on the definition from 2% to 6%) of land [1–3] but play an important role by providing ecosystem goods and services at various spatial scales.

At the global scale, wetlands accumulate a significant amount of organic carbon [4–6] and regulate the regional climate [7]. At the watershed scale, wetlands fulfil an important hydrological function by controlling different components of the water balance such as snow retention [8], flood retention [9–11], evapotranspiration [12], and interception [13,14]. At the local scale, wetlands provide food, remove water pollution, retain water (hence, controlling floods), stop sediments, and much more [15]. Despite their importance, human activities, e.g., drainage for the farming and cultivation of crops, caused major changes in these vulnerable ecosystems [16]. Ongoing climate changes will make future efforts to restore or maintain valuable wetland functions more complex [17].

One of the most serious threats to maintaining the wetlands services and functions is drought occurring more frequently and severely in recent years [18,19]. Various types of drought can be distinguished: Meteorological (defined in terms of the magnitude of a precipitation shortfall and the duration of this shortfall event), agricultural (defined as a deficit of soil moisture), hydrological (defined as the lowering of surface and subsurface water supply), and socioeconomic (defined in terms of socioeconomic losses caused by other types of drought) [20]. Agricultural drought, through the lowering of groundwater level on wetlands, leads to changes in the physical soil characteristics and mineralization of organic matter by enhanced rates of microbial decomposition [21]. These processes cause the reduction of peat porosity [22,23], which consequently results in the process of the disappearance of plant species, characteristic for wetland habitats (e.g., *Valeriana simplicifolia*, *Epipactis palustris*, *Eleocharis quinqueflora*, *Limprichtia cossonii*, and *Campylium stellatum* for studied alkaline fens – Natura 2000 habitat code 7230; and *Scheuchzerietalia palustris* order, *Caricetalia nigrae* order, *Scheuchzerio-Caricetea* class: *Comarum palustre* and *Eriophorum angustifolium* for studied transition mires and quaking bogs – Natura 2000 habitat code 7140) [24]. These processes also consequently result in the reduction in the wetland's water capacity, the increase in air content in the soil and its temperature, and, finally, in higher $CO_2$ emissions to the atmosphere [25,26]. Therefore, the monitoring of drought and its consequences are remarkably important in wetland areas. In agriculture, due to high market demands, a significant number of meteorological and remote sensing drought indicators were developed [27–29]. Unfortunately, due to the specificity of wetlands, their monitoring by both traditional and remote sensing techniques is challenging [30] and, therefore, they are less recognized in terms of water stress. In order to assess the occurrence and intensity of droughts in agricultural areas [31], the following indices are used, among others: Soil moisture index [32] or crop water stress index (CWSI) [33]. The CWSI found wide application possibilities [34], among others, in determining the needs of the irrigation of agricultural and horticultural crops [35,36]. The wide range of CWSI applications in addition to agricultural crops includes its use in assessing the water status of trees: Fruit orchards [37,38], olive [39], and almond trees [40]. The advantage of CWSI application is it requires only basic meteorological parameters (air temperature and relative humidity), canopy temperature, and limiting cases derived empirically [41] or theoretically [42]. Hence, this paper describes the potential use of the well-known and widely used health monitoring of agriculture crop CWSI for determining water stress for major European wetland habitats: Alkaline fen, transition mires, and quaking bogs. In order to achieve this goal, non-water-stressed baselines (NWSBs) for these habitats were developed based on field measurements (section 2.2.2). The CWSI was calculated on two test areas (one for each habitat) based on the unmanned aerial system (UAS) thermal orthophotomosaic (section 2.3.), and field campaigns during UAS acquisitions were conducted. Soil moisture, a fraction of absorbed photosynthetically active radiation (fAPAR), and chlorophyll content index (CCI) were measured. Besides field measurements, two meteorological drought indicators (SPI - standardized precipitation index, and SCWB - standardized climatic water balance) were calculated. Finally, field measurement results and meteorological drought indicator values were compared to the CWSI to analyze whether the CWSI can be used as water stress or drought indicators in wetlands.

## 2. Materials and Methods

### 2.1. Study Sites

The research was conducted on two semi-natural habitats of Natura 2000 (Figure 1). The research of habitat 7230 (alkaline fens) was carried out in the Biebrza National Park area located in north-eastern Poland during the growing season of 2016. The research of habitat 7140 (transition mires and quaking bogs) was carried out in the Janów Forest Landscape Park area in south-eastern Poland during the growing season of 2017.

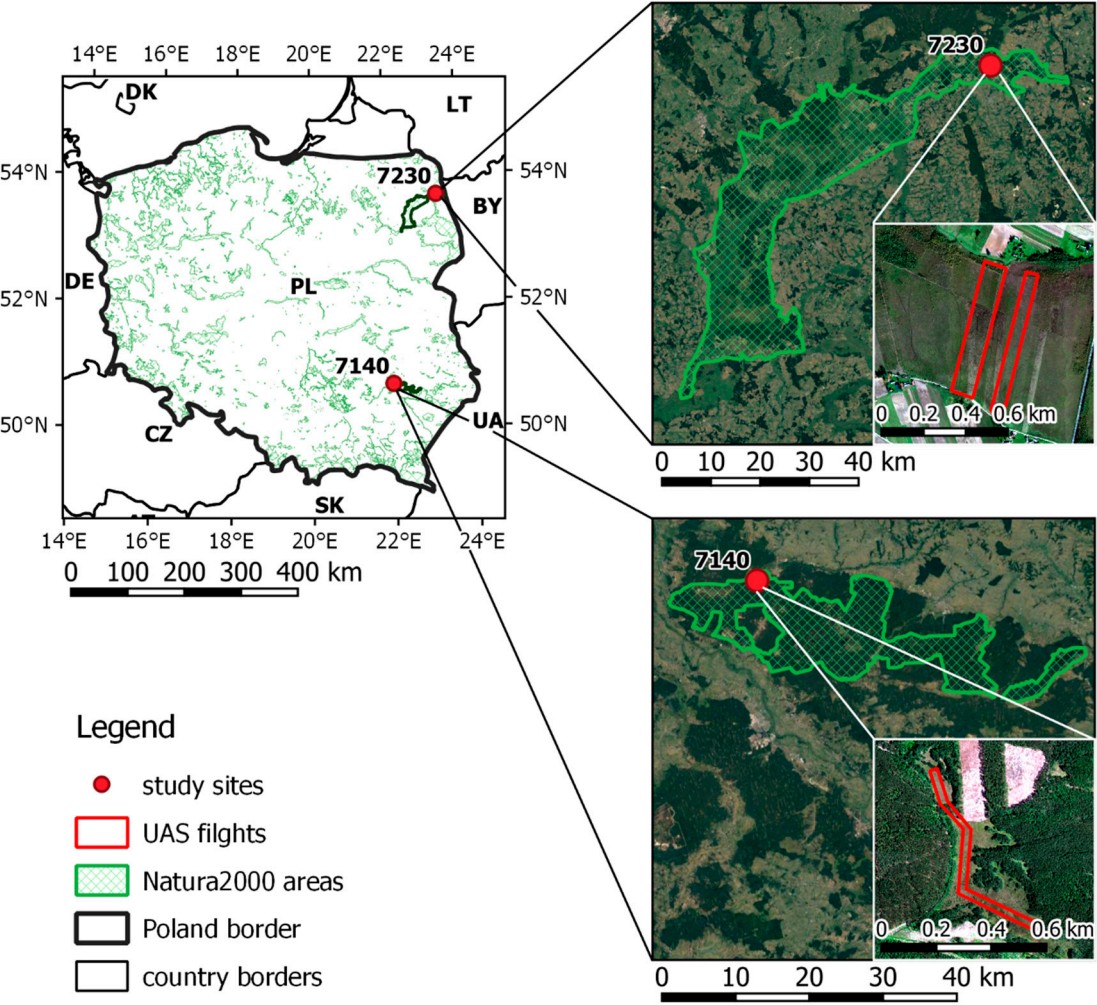

**Figure 1.** Study sites location on Natura 2000 area map of Poland.

### 2.1.1. The Biebrza National Park

Research for habitat 7230 was carried out in the Upper Biebrza Valley in the area of the Biebrza National Park (Figure 1). The Upper Biebrza Valley is a north-eastern part of the Biebrza Proglacial Stream Valley, which, together with sandurs adjoining from the north, constitutes the largest lowering of north-eastern Poland with a length of approx. 150 km and an area exceeding 2600 km². The Proglacial Stream Valley constitutes the largest complex of natural peatlands in Central Europe. In relation to the adjacent areas, it has different air thermal and humidity characteristics. This is due to the overlap of climate features associated with the vast areas of peatlands and the main features of north-eastern Poland's climate, with an average annual temperature of 6–7 °C (1971–2000) [43]. The average annual precipitation from the multiannual term of 1971–2000 is 550–600 mm in this area [43], while the duration of snow cover lasts from 90 to 110 days. The research was carried out in the area near the Szuszalewo village characterized by extensively exploited alkaline peatlands, currently one of the most endangered natural habitats in Europe [44]. In the case of the alkaline peatlands, the groundwater level usually remains at or slightly above the ground surface in optimal conditions. The habitat is characterized by low fertility and a high content of calcium ions. The dominant vegetation

is moss-sedge communities (*Scheuchzerio-Caricetea nigrae* community), characterized by a large floral richness with a large share of rare and protected species, such as *Liparis loeselii*, or plant species from *Menyantho trifoliatae-Sphagnetum teretis*, *Caricion davallianae*, *Caricetum rostratae*, and *Caricetum paniceo-lepidocarpae* communities. A very important group of plants within habitat 7230 are mosses, part of which are glacial relics. Changes in groundwater level, as well as the succession toward forest and shrub communities, pose the greatest threat to the habitat.

### 2.1.2. The Janów Forest Landscape Park

Research for habitat 7140 was carried out in the Janów Forest Landscape Park, which, together with the neighboring Solska Wilderness, is one of the largest forest complexes in Poland (Figure 1). According to the physiographic division of Poland [45], this area is located in the mesoregion of the Biłgoraj Plain called Puszczańska, which is part of the Sandomierz Basin. The studied area lies in the Lublin district [46], which is relatively warm with an average annual air temperature of 7–8 °C (1971–2000). The average annual precipitation amounts to 550–600 mm (1971–2000) [43], while the time of snow cover lasts from 80 to 90 days. The area of the Janów Forest Landscape Park mainly consists of forest habitats, where the largest area is occupied by mixed wet coniferous forest and fresh coniferous forest. Aquatic-peat and aquatic communities are characterized by the greatest floristic richness. Habitat 7140 (transition mires and quaking bogs), within which the research was carried out, constitutes approximately 400 ha of the area of the Janów Forest Landscape Park. They are mainly peat-filled no-runoff land lowerings with a groundwater level arranged in optimal conditions at or slightly above the ground surface. Vegetation primarily consists of a moss and herbaceous layer, where the moss layer is composed mainly of *Sphagnum* moss and *Bryopsida*. Usually, one or two species of plants predominate in the habitat. In the Janów Forest Landscape Park, there are species of plants from *Eriophoro angustifolli-Sphagnetum recurvii*, *Caricetum rostratae*, *Caricetum lasiocarpae*, *Eriophorum vaginatum-Sphagnum fallax*, or *Rhynchosporetum albae* communities. Changes in hydrological conditions, as well as the encroachment of trees and shrubs into open wetland areas in the conditions of lowering the groundwater level, pose the greatest threat to habitat 7140.

### 2.2. CWSI

Calculation of the CWSI requires the NWSB for investigated habitats. In this section, the formula for CWSI calculation is first presented. Secondly, the methodology of the NWSB derivation is described.

### 2.2.1. Formulation

The CWSI was calculated according to the following equation [41,42]:

$$\text{CWSI} = (dT_m - dT_{LL})/(dT_{UL} - dT_{LL}), \tag{1}$$

where dT is the temperature difference between canopy ($T_c$) and air ($T_a$), and the subscripts m, LL, and UL respectively refer to the measured difference and the lower (non-water-stressed) and upper limit of dT. The upper and lower limits of dT can be estimated based on the empirical [41] or theoretical approach [42]. In this study, an empirical approach was used. It is based on the assumption that $dT_{LL}$ is linearly related to the vapor pressure deficit (VPD) for a non-water-stressed habitat patch under specific climatic conditions – this relation is further referred to as the NWSB. Similarly, there is a linear relation between $dT_{UL}$ and the vapor pressure gradient (VPG) for the same habitat patch when its transpiration is halted due to severe water stress. The upper and lower limits were calculated according to the following equations:

$$dt_{LL} = m \cdot VPD + b, \tag{2}$$

$$dt_{UL} = m \cdot VPG + b, \tag{3}$$

where "m" and "b" respectively refer to slope and intercept of the NWSB. The VPG was estimated according to [47] based on the saturation vapor pressure (E) at the instantaneous air temperature and E at the instantaneous air temperature elevated by NWSB's coefficient "b" as their difference:

$$VPG = E(T_a)-E(T_a+b). \tag{4}$$

In this study, the CWSI was calculated based on UAS $T_c$ acquisitions (section 2.3.) conducted simultaneously with the basic meteorological measurements (air temperature and relative humidity). Meteorological measurements were made using a HOBO U23 Pro v2 Temperature/Relative Humidity Data Logger sensor (Onset®, USA).

### 2.2.2. The NWSB derivation

The NWSB was developed for both investigated habitats based on field and laboratory measurements. In order to develop the NWSB $T_a$, $T_c$ and VPD need to be measured or calculated. The measurements should be performed when a given habitat is in non-water-stressed conditions under clear sky conditions [47]. Hence, field measurements of canopy temperature, air temperature, and relative humidity (RH) were conducted on dates after a longer period of optimal water conditions (dates were selected based on groundwater level and precipitation monitoring). Based on $T_a$ and RH, the VPD was calculated according to the following formula [48]:

$$E = 0.6108 \cdot exp[(17.27 \cdot T_a)/(T_a+237.4)], \tag{5}$$

where E is the saturation vapor pressure at the air temperature $T_a$ [kPa].

$$E = e \cdot RH, \tag{6}$$

where e is the actual vapor pressure [kPa].

$$VPD = E-e. \tag{7}$$

For NWSB determination, $T_c$ was measured using a non-contact handheld OMEGA Os151-usb thermometer (OMEGA Engineering, INC), and $T_a$ and RH were measured using the HOBO U23 Pro v2 Temperature/Relative Humidity Data Logger sensor (Onset®, USA). All measurements were done with 30 min intervals. Only measurements under a clear sky (cloudless) were used in the further determination of the NWSB.

Coefficients a and m of the NWSB (Equation (2)) were fitted based on field measurements using the least-squares method. For the fitted lines, the coefficient of determination and its p-value were calculated.

### 2.3. UAS Data Capture and Thermal Orthophotomosaic Preparation

The land surface temperature (LST) was recorded over research transects (Figure 1) using a thermal camera installed on the UAS. The research transects were selected based on their availability for UAS (enough space for take-off and landing) and plant coverage. In the selected areas in the growing season, plants are dense enough to assume that LST is equal to canopy temperature. The UAS acquisitions were conducted in 3 field campaigns in the Biebrza National Park and in 5 field campaigns in the Janów Landscape Park (Table 1). For the purposes of this research, the UAS was constructed based on the DJI S1000+ frame (DJI, China) equipped with PIXHAWK autopilot (3D Robotics, USA) with a built-in inertial measurement unit (IMU) system. GNSS navigation data were collected using the Tersus-GNSS Precis BX306 GNSS device (Tersus GNSS China, China), enabling dual-frequency signal recording in both GPS and GLONASS systems. The land surface temperature (LST) was recorded using an Optris PI640 radiometric camera (Optris GmbH, Germany) mounted on a two-axis stabilizing system. Additionally, in order to increase image interpretation possibilities, RGB images were recorded using a UAS DJI Phantom 3 Professional platform (DJI, China). For both, thermal and RGB dataset ground control points (GCPs) were recorded.

**Table 1.** Acquisition dates and time, the number of transects, flights per transect, and vapor pressure deficit (VPD) values during flights for both research areas.

| Area | Date | Number of transects | Flights per transect | Time of flights | Minimum VPD (kPa) | Maximum VPD (kPa) |
|---|---|---|---|---|---|---|
| Biebrza National Park (7230). | 24.07.2016 | 2 | 3 | 11:00–3:30 | 1.22 | 1.41 |
| | 07.09.2016 | | 2 | 12:00–2:00 | 1.05 | 1.13 |
| | 18.09.2016 | | 2 | 11:00–2:00 | 1.21 | 1.43 |
| Janów Forest Landscape Park (7140) | 14.07.2017 | 1 | 3 | 11:30–2:00 | 0.85 | 1.01 |
| | 01.08.2017 | | 3 | 11:40–3:00 | 1.25 | 3.43 |
| | 19.08.2017 | | 1 | 12:30 | 2.66 | 2.66 |
| | 30.08.2017 | | 3 | 11:50–2:00 | 1.28 | 1.59 |
| | 09.09.2017 | | 3 | 11:00–12:30 | 0.99 | 1.36 |

Flights over the research transects were carried out autonomously along the planned routes, which ensured repeatability of the area extent range that was acquired during subsequent flights. The level of side lap coverage was maintained at a value of at least 75%. The fixed flight parameters allowed a terrain resolution not worse than 10 cm to be obtained. Air operations were carried out in possibly stable weather conditions (wind speed below 5 m/s, clear sky, and no rapid change in air temperature and humidity). When meteorological conditions allowed more flights to be conducted, up to 3 flights per transect were conducted.

The coordinates of the projection centers of individual radiometric images were determined based on the GNSS on-board system recorded data. Then, the data set (the image–GPS position pair) was processed in Photoscan software (Agisoft LLC, Russia). The photogrammetric process consisted of generating a sparse point cloud, controlling the initially obtained model, creating a dense cloud of points, creating a digital surface model (DSM), and finally constructing an orthophotomosaic and exporting it to the TIFF format.

For improving the geometrical accuracy of the thermal orthophotomosaic additional RGB UAS, reference data were captured and the GCP network was established. A color (RGB) orthophotomosaic was prepared for each of the research transects. The selected flight parameters allowed reference data to be obtained with a terrain resolution higher than 5 cm. The coordinates of each GCP were measured using the GNSS (Global Navigation Satellite Systems) RTK (Real Time Kinematic) technique using GNSS receiver Topcon GRS-1 (Topcon, Japan) in real-time corrections from the TPI NetPRO network [49]. The GCPs were used during the photogrammetric process to increase the internal coherence and external accuracy of the resulting RGB orthophotomosaic. The obtained error of fitting on the matrix points did not exceed 3 pixels (regarding the resolution of the orthophotomosaic at the level of 5 cm).

Next, the LST orthophotomosaic was subjected to final geometric rectification in the QGIS software based on characteristic points identifiable on the RGB orthophotomosaic. First, given that the characteristic point was identified on the LST image, it was then fitted into corresponding coordinates based on the RGB orthophotomosaic. At least 20 points per study area evenly distributed over the image were used.

*2.4. Meteorological Drought Indices*

Apart from the CWSI, two meteorological drought indices, namely, SPI and SCWB, were calculated with meteorological data (air temperature, precipitation, relative humidity, wind speed) gathered by the Institute of Meteorology and Water Management, National Research Institute (IMGW-PIB) [50]. The SPI and SCWB were selected as the most valuable and reliable meteorological drought indices. Meteorological drought indices were calculated to test which type of drought can be monitored using the CWSI.

The SPI was calculated according to the original methodology proposed by [51]. The SPI was calculated as the difference of precipitation from the mean for a specified time period divided by the standard deviation calculated from past records. For the calculation of SPI precipitation, data from the last 30 years were used.

The SCWB is a standardized deviation of climatic water balance values in a given period by the mean long-term value of this period [52]. Climatic water balance is calculated as the difference between total precipitation and the reference evapotranspiration. For SCWB calculation, the reference evapotranspiration was calculated according to FAO (Food and Agriculture Organization of the United Nations) [48]. Daily values of SCWB for a 30 year data series were calculated.

Both indices were calculated for a period of 3 months as a moving average using a dedicated package in the R environment [53].

Classifications of the SPI and SCWB values and adopted meteorological drought categories are presented in Table 2.

**Table 2.** Classification of meteorological drought according to standardized precipitation index (SPI) and standardized climatic water balance (SCWB) values.

| Meteorological drought category | SPI values [51] | SCWB values [52] |
|---|---|---|
| mild drought | 0 to –0.99 | 0.50 to –0.99 |
| moderate drought | –1.00 to –1.49 | |
| severe drought | –1.50 to –1.99 | |
| extreme drought | ≤–2.00 | |

*2.5. Biophysical Parameters, Soil Moisture, and Groundwater Level*

The measurements of CCI and fAPAR, as well as the moisture content of the surface soil layer, took place on the dates of the acquisition of UAS data. These parameters describe the habitat condition and water stress of plants. A total of 45 in the Biebrza National Park and 23 in the Janów Forest Landscape Park plots (Figure 3 and 4) with dimensions of 1 x 1 m were established, in which the above-mentioned measurements were carried out. In all plots, plant composition was also noted. Before using these points in statistical analyses, each one was verified on an RGB orthophotomosaic, to exclude disrupted (e.g., by mowing, water, and shadow occurrence) points. In addition, points with the peculiar plant composition were excluded.

Droughts affect the vegetation capacity of intercepting solar radiation, which can be described by the fAPAR. The fAPAR determines the fraction of solar radiation that has been absorbed by plants during the photosynthesis. It is expressed by the ratio values of photosynthetically active accumulated radiation to total radiation reaching the surface of plants. The parameter allows the condition of plants and their productivity and the effects of water deficit to be assessed, which are different according to the plant's growth stage. However, a reduction in the intercepted radiation (and, therefore, in fAPAR) is always a consequence of droughts [54,55]. Measurements of the fAPAR were carried out using the SunScan Canopy Analysis System SS1 system (Delta-T Devices, Ltd., UK). Within each research plot, a total of 10 measurements of fAPAR were carried out and subsequently averaged.

The chlorophyll content is one of the major factors influencing photosynthetic capacity. Due to that, changes in chlorophyll content in tissues in a plant under drought stress have been observed in different species. Its intensity depends on the stress rate and duration [56]. Many authors proved that the chlorophyll content of a leaf can be a reliable indicator of photosynthetic capability [57,58]. Fotovat et al. [59] found that by exerting severe drought stress on wheat, the chlorophyll content of a leaf significantly decreased. The CCI measurements were carried out using a CCM200 Chlorophyll

Meter (Opti-Sciences, Inc., USA), which uses the transmittance measured in two wavelength ranges of 653 and 931 nm to estimate the relative chlorophyll content in leaves. The CCI value is directly proportional to the actual concentration of chlorophyll in the leaves. The index made it possible to observe changes in the physiological state of dominant plant species under the influence of changing soil moisture that may have caused water stress. CCI measurements were carried out for 10 individuals of each of the 1–3 dominant species found on the research plot in triplicate (on three different leaves), which were then averaged.

Measurements of moisture in the surface soil layer (0–10 cm) were carried out at each research plot in triplicate using a time domain reflectometer (TDR) [60]. The obtained results representing dielectric constant (Ka) were averaged to a single value for each plot. The conversion of Ka values into the value of the soil moisture content requires a calibration equation. Authors used their own calibration equations developed in laboratory tests for the soils of 7230 and 7140 habitats in accordance with the methodology presented by [61,62].

Additionally, in each study area, piezometers were installed (3 in the Biebrza National Park and 2 in the Janów Forest Landscape Park). Piezometers were equipped with automatic Levelogger® Edge (Solinst Canada Ltd.) sensors, which recorded data of the groundwater level every 3 h. Groundwater level data were averaged for 24 h for each study area.

### 2.6. Data Analysis

The first step after the NWSB determination (Section 3.1.) and preparing thermal orthophotomosaic (Section 2.3.) was the calculation of the CWSI (Equation (1)) for the whole UAS flight extent. Then, results were masked using habitat extent maps (only areas within selected habitats were analyzed). These results allowed the CWSI to be analyzed spatially and temporally (Section 3.2.). In the second step, CWSI values were compared to precipitation, temperature, groundwater level, and meteorological drought indices (Section 3.3.). In the last step, field measurements of biophysical parameters and soil moisture were used to analyze the correlation between them and CWSI values (Section 3.4.). The analysis was performed on all points from both areas. The correlations were analyzed using Pearson's correlation coefficient in the R environment [63].

## 3. Results

### 3.1. NWSB

Field and laboratory measurements of $T_c$, $T_a$, and RH allowed the NWSB to be determined for both habitats (Figure 2). The fitted lines have satisfactory (for habitat 7140) and high (for habitat 7230) determination coefficients (Table 3.). Measurements for habitat 7140 were done during the day with VPD values ranging from 0.73 to 1.89 kPa and observed the temperature difference between the canopy and air ranging from 2.1 °C to 12.9 °C. To analyze the applicability to wider meteorological conditions, measurements were repeated in the laboratory on a representative habitat sample. For habitat 7230, VPD values ranged from 1.03 to 1.87 kPa and temperature differences ranged from –0.9 °C to 11.0 °C. The VPD values during NWSB determination in most cases correspond to VPD values during flights (Table 1).

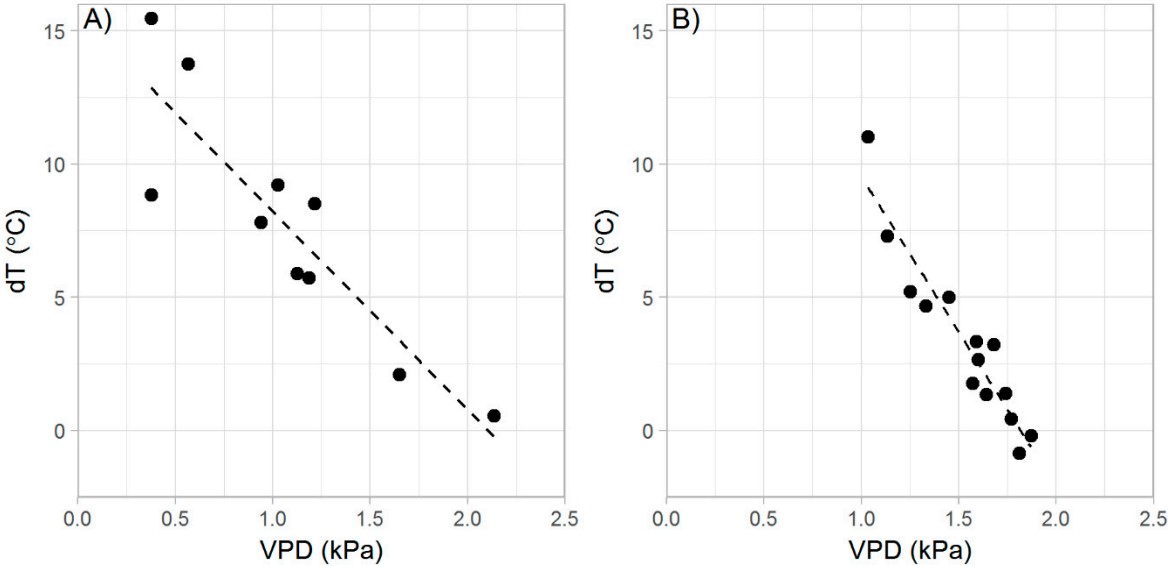

**Figure 2.** The non-water-stressed baseline (NWSB) for (**A**) habitat 7140 and (**B**) habitat 7230.

**Table 3.** The NWSB statistics and parameter values for each habitat.

| Area | m | b | $R^2$ | p-value | Number of measurements |
|---|---|---|---|---|---|
| Janów Forest Landscape Park (habitat 7140) | –7.4 | 16.3 | 0.64 | <0.01 | 10 |
| Biebrza National Park (habitat 7230) | –11.6 | 21.3 | 0.89 | <0.01 | 14 |

*3.2. CWSI*

The results of CWSI calculations are presented on maps within the flight extent (Figure 3 and 4) and on boxplots for each research area and transect (Figure 5). The significance of median differences in CWSI values between particular field campaigns in both areas was checked using the Kruskal–Wallis test in the R environment [63]. For both areas, differences in CWSI values were significant at the level of $p < 0.05$.

The CWSI calculation results for the Biebrza National Park (Figure 3 and 5) showed the lowest values (median equal to –0.012) during the first measurement campaign (24.07.2016). In the next campaign on 07.09.2016, the median value increased to –0.008. In this campaign, the range of CWSI values was higher than in the first one. Further (in the last campaign), the CWSI achieved the highest values (median equal to 0.034). The spatial distribution of CWSI values in the second and third campaign showed higher CWSI values in the northern part of the two studied transects and the western part of the west transect. This part of the transect was mowed just before the second field campaign. Hence, the results showed a visible spatial pattern of CWSI and the consequent likelihood that vegetation is subject to water stress in this area.

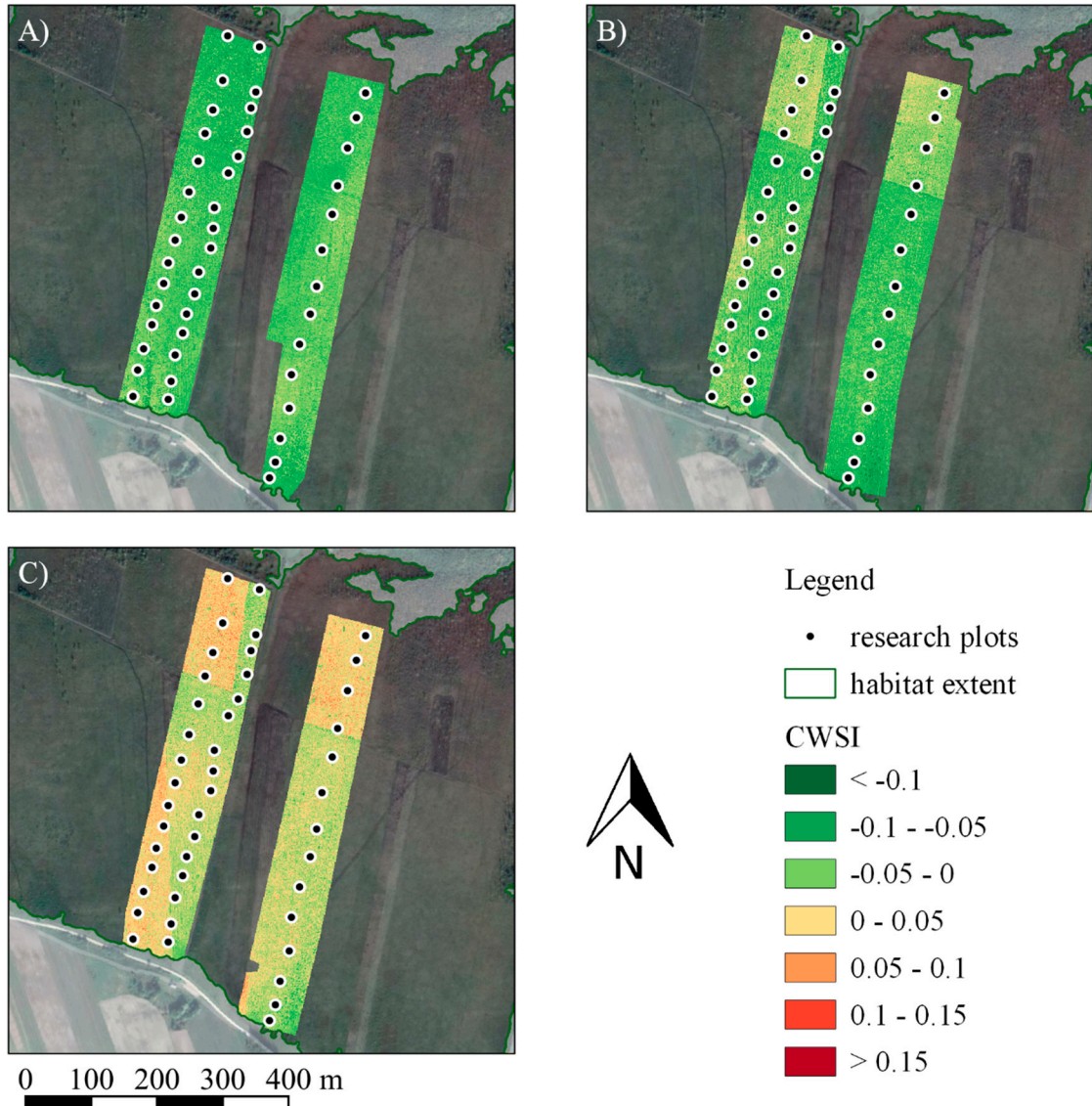

**Figure 3.** The spatial distribution of crop water stress index (CWSI) with the location of research plots in the Biebrza National Park for 3 unmanned aerial system (UAS) campaigns in 2016: (**A**) 24 July, (**B**) 7 September, and (**C**) 15 September.

The calculation results for the area of Janów Forest Landscape Park (Figure 4 and 5) show low values (median equal to 0.018) of CWSI during the first measurement campaign (14.07.2017). Then, the CWSI reaches its maximum (median equal to 0.165) during the second measurement campaign (1.08.2017). Further, the values of the CWSI gradually decrease, reaching a minimum (median equal to –0.046) during the last measurement campaign (09.09.2017). The spatial distribution of CWSI values in the first four campaigns showed the lowest values of CWSI in the southern part of the flight area (Figure 5). In the fifth campaign, nearly all of the research area was characterized by CWSI values below 0 and the southern part did not stand out significantly. Apart from the temporal pattern of water stress, the CWSI also showed spatial variability and indicated parts of natural habitats that were less vulnerable to water stress caused by droughts. The field assessment confirmed the better overall condition of plants in locations where small ponds were present.

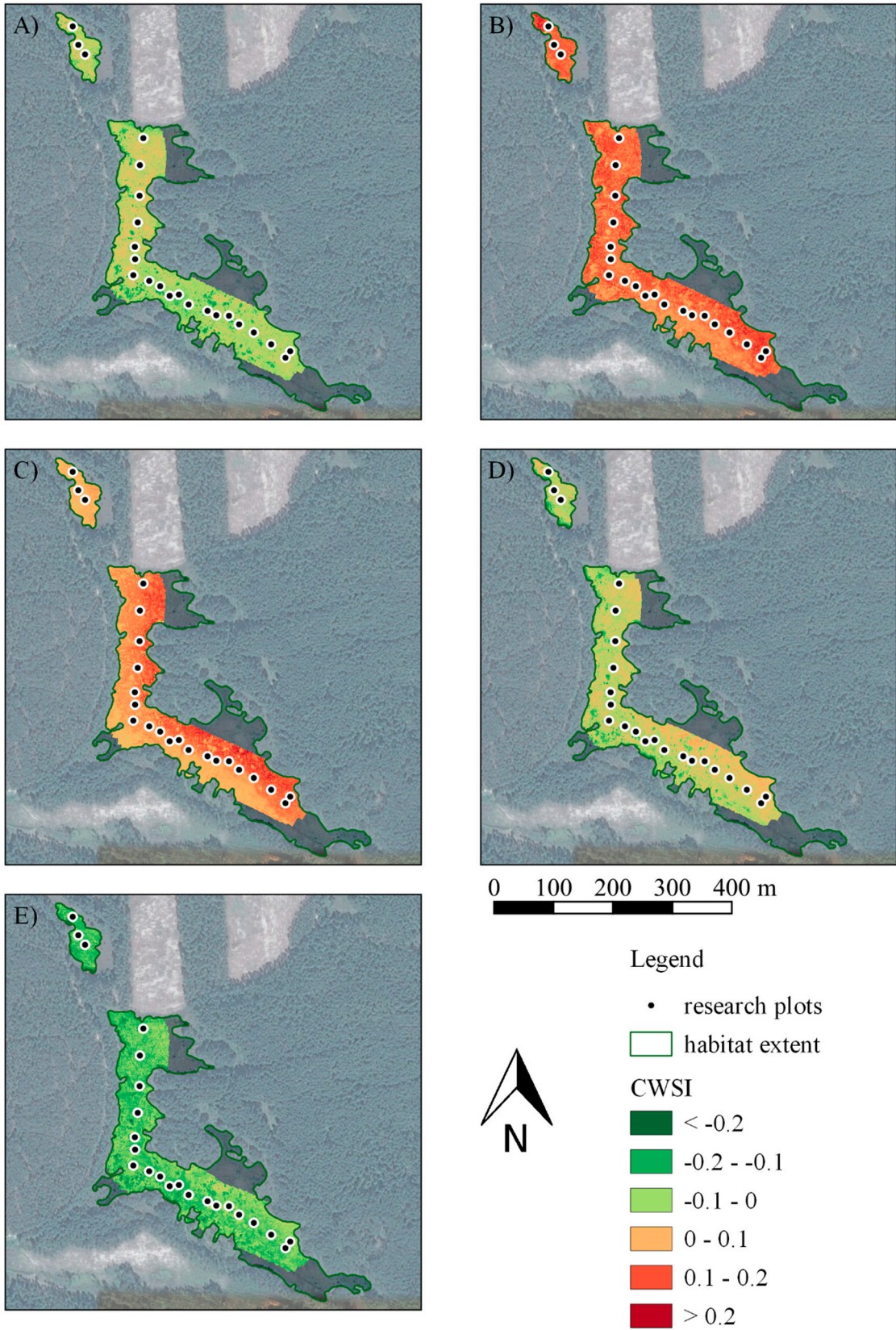

**Figure 4.** The spatial distribution of CWSI with the location of research plots in the Janów Forest Landscape Park for 5 UAS campaigns in 2017: (**A**) 14 July, (**B**) 1 August, (**C**) 19 August, (**D**) 30 August, and (**E**) 9 September.

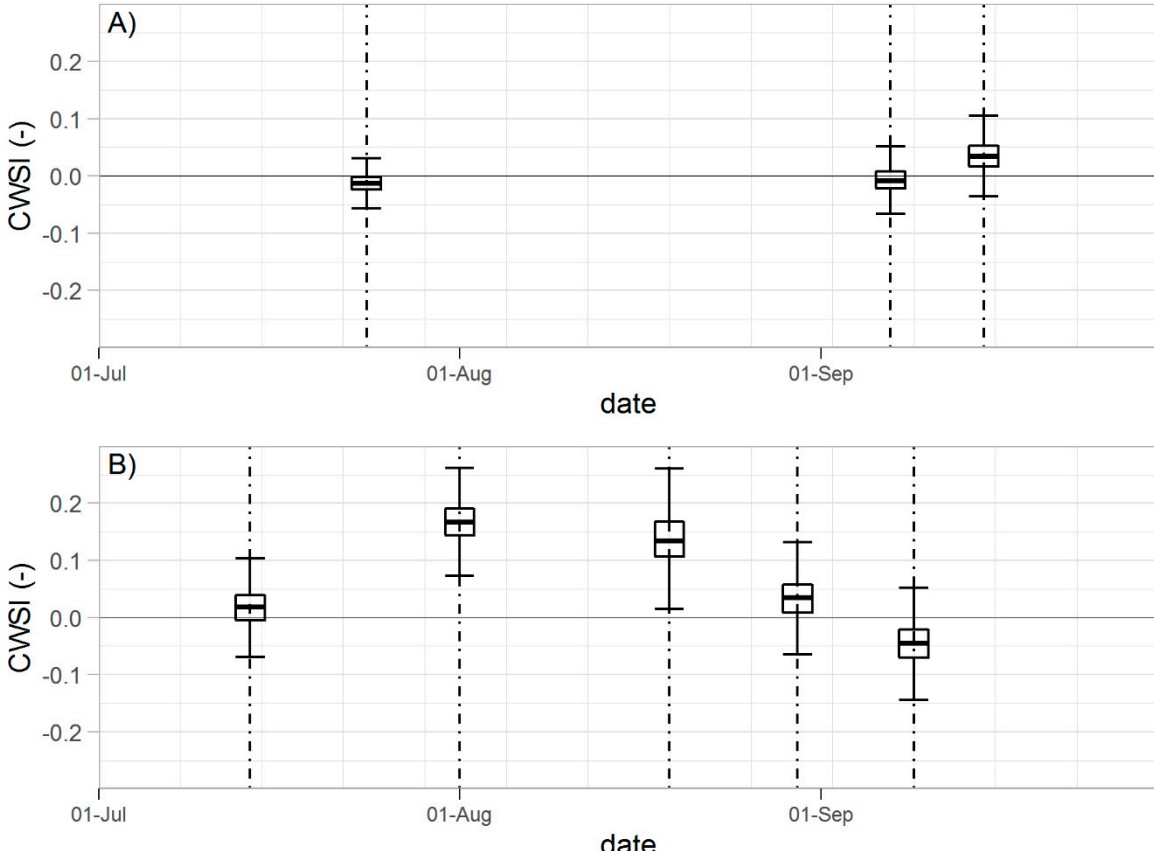

**Figure 5.** The CWSI values distribution for (**A**) The Biebrza National Park and (**B**) the Janów Forest Landscape Park. The boxplot shows the median, the lower and upper hinges correspond to the first and third quartiles, respectively, the upper and lower whiskers are the 1.5·interquartile range, respectively, and the dash-dotted vertical lines indicate UAS flight campaigns (Table 1).

*3.3. Meteorological Parameters and Drought Indices*

The course of hydrometeorological data in the Biebrza National Park (Figure 6A) corresponds to temporal changes in the CWSI observed within the flight extent (Figure 5A). In the first field campaign (24.07.2016), in the major part of the research area, CWSI values were below 0 (indicating the occurrence of the appropriate habitat conditions through the research period). During this campaign, the groundwater level (determining the availability of water for plants) was the highest (–10 cm), due to the earlier long period with precipitation, and remained higher than –15 cm until the end of August, where it started to decrease until the end of the growing season in September. However, we observed a slight increase at the beginning of September, which was caused by two days with high precipitation. The fluctuation of the CWSI values showed increases in the second (07.09.2016) and third (18.09.2016) campaign, representing less water availability.

The values of hydrometeorological parameters in the Janów Forest Landscape Park (Figure 6B) correspond to the temporal change in the CWSI in a flight extent (Figure 5B) as it was observed in the Biebrza National Park. In the first field campaign (14.07.2017), the CWSI values were above 0 in the major part of the research area (indicating the occurrence of average hydrological conditions through the research period). During this campaign, the groundwater level was equal to –21 cm. In the second field campaign (01.08.2017), groundwater level decreased to –30 cm and the CWSI reached its maximum. In the third field campaign (19.08.2017), the CWSI indicated the occurrence of better habitat conditions than in the second one (01.08.2017), although the groundwater level reached the lowest value (–58 cm). The cause was higher air temperature (hence, vapor pressure deficit as well) in the second campaign, and this parameter plays a crucial role in CWSI determination. After the third field campaign (19.08.2017), the groundwater level started to rise and the CWSI values correspond to this change (air temperature in the last three campaigns was similar). In the fourth field

campaign (30.08.2017), the groundwater level was equal to –33 cm and the CWSI was slightly higher than in the first field campaign. In the last field campaign (09.09.2017), when the groundwater level reached –13 cm, the CWSI values were the lowest (in major part lower than 0) through the whole research period.

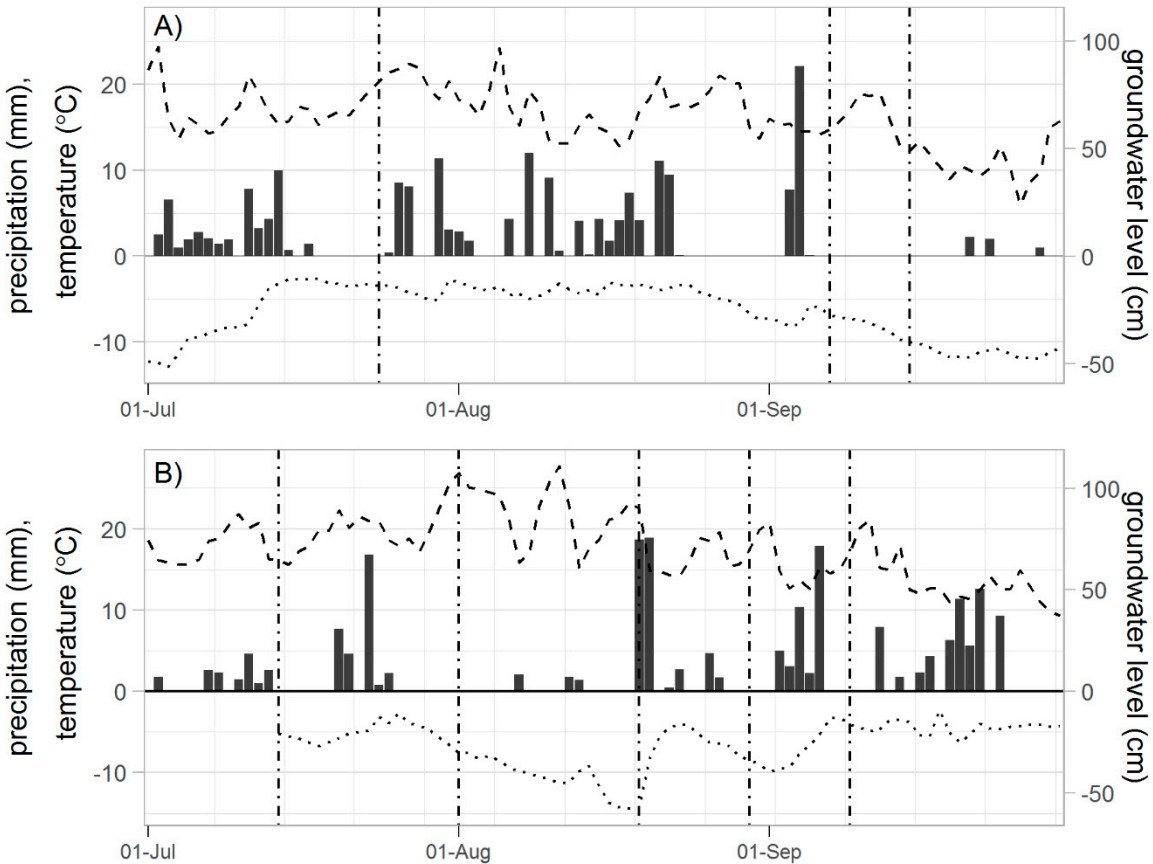

**Figure 6.** Precipitation (black bars), temperature (dashed lines), and groundwater level (dotted lines) for (**A**) the Biebrza National Park and (**B**) the Janów Forest Landscape Park. Vertical dash-dotted lines indicate UAS flight campaigns (Table 1).

The SPI values for the Biebrza National Park (Figure 7A) in all campaigns were slightly above 0, indicating non-meteorological drought conditions. The SPI values were equal to 0.062, 0.039, and 0.085 for the first, second, and third campaigns, respectively. The SCWB values for the Biebrza National Park (Figure 7A) in all campaigns indicated mild drought with values equal to –0.484, –0.346, and –0.086 for the first, second, and third campaign, respectively.

In the Janów Forest Landscape Park in all campaigns (besides the third one), the SPI had low positive values (Figure 7B), indicating non-meteorological drought conditions. In the third campaign, the SPI value increased significantly and was equal to 2.275. The SCWB in the first, fourth, and fifth campaigns was equal to –0.285, –0.579, and –0.844 respectively. These values indicate mild drought conditions. In the second campaign, the SCWB value dropped significantly and was equal to –1.989, indicating severe drought. Similar to the SPI, in the third campaign, the SCWB increased significantly and was equal to 2.293. The lowest values of both indices were observed in the second campaign (01.08) when the highest values of CWSI (Figure 5B) were observed. In the third campaign, high values of CWSI (Figure 5B) were observed, but both meteorological drought indices had the highest values and did not correspond to CWSI indications. The precipitation that occurred on the 19th and 20th of August was 37.6 mm and affected the SPI and SCWB results in the third campaign. It should be noted that the CWSI in this campaign (19.08.2017) was calculated based on the flight done a few hours before precipitation in more water-stressed conditions.

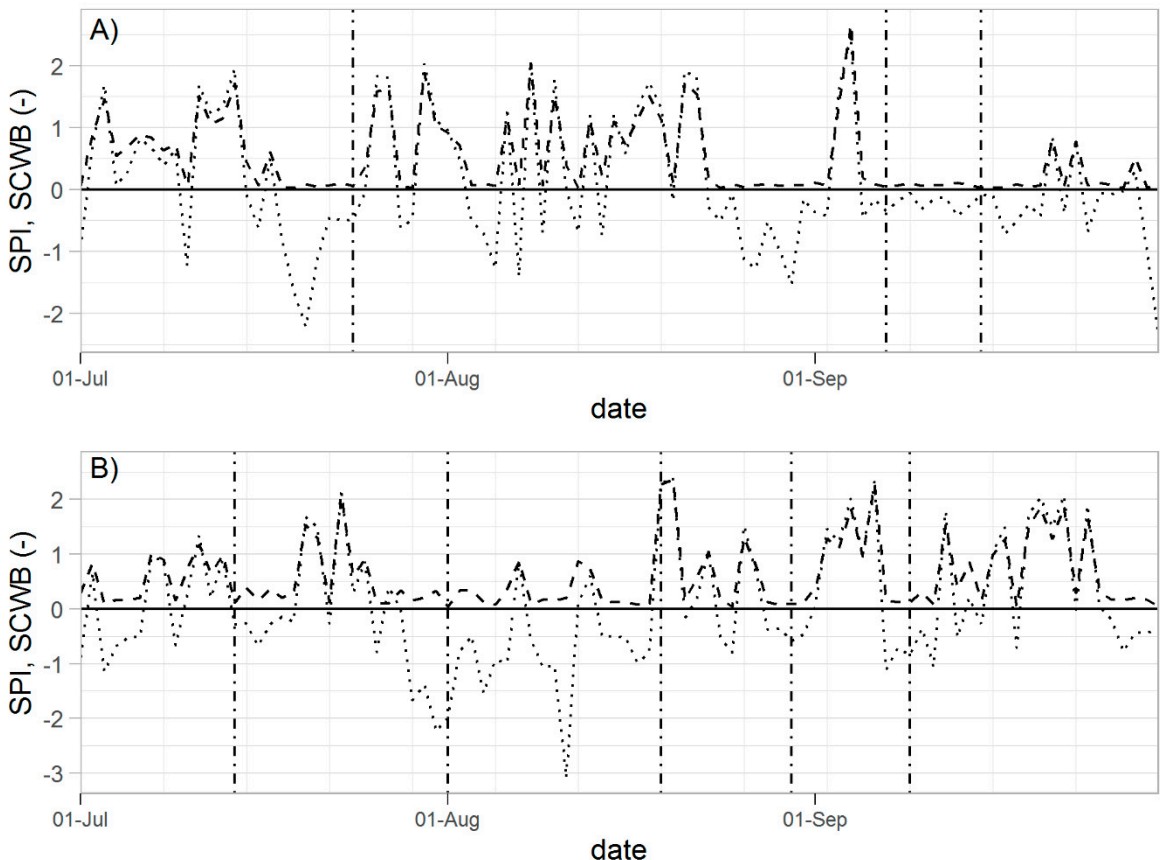

**Figure 7.** Drought indices: SPI (dashed lines) and SCWB (dotted lines) for (**A**) the Biebrza National Park in 2016 and (**B**) the Janów Forest Landscape Park in 2017. Vertical dash-dotted black lines indicate UAS flight campaigns (Table 1).

*3.4. Correlation of CWSI and Field Measurements*

All CWSI values from all research plots were plotted against field measurements of soil moisture (Figure 8A), CCI (Figure 8B), and fAPAR (Figure 8C). The number of points used for analysis, Pearson's correlation coefficient, and its p-value for the plotted correlation are presented in Table 4.

As expected, CWSI is inversely proportional for biophysical parameters values. The absolute value of correlation coefficients for soil moisture and fAPAR (Table 4) is higher than 0.5, showing a strong correlation between this parameter and CWSI. The CCI correlation coefficient (Table 4) shows a weaker correlation than the two other parameters. All three relations are statistically significant. Observed values of Pearson's correlation coefficient indicate that the CWSI can be used as a water stress indicator for wetland ecosystems. High CWSI values (indicating water stress) correspond to low values of the measured parameters (also indicating water stress).

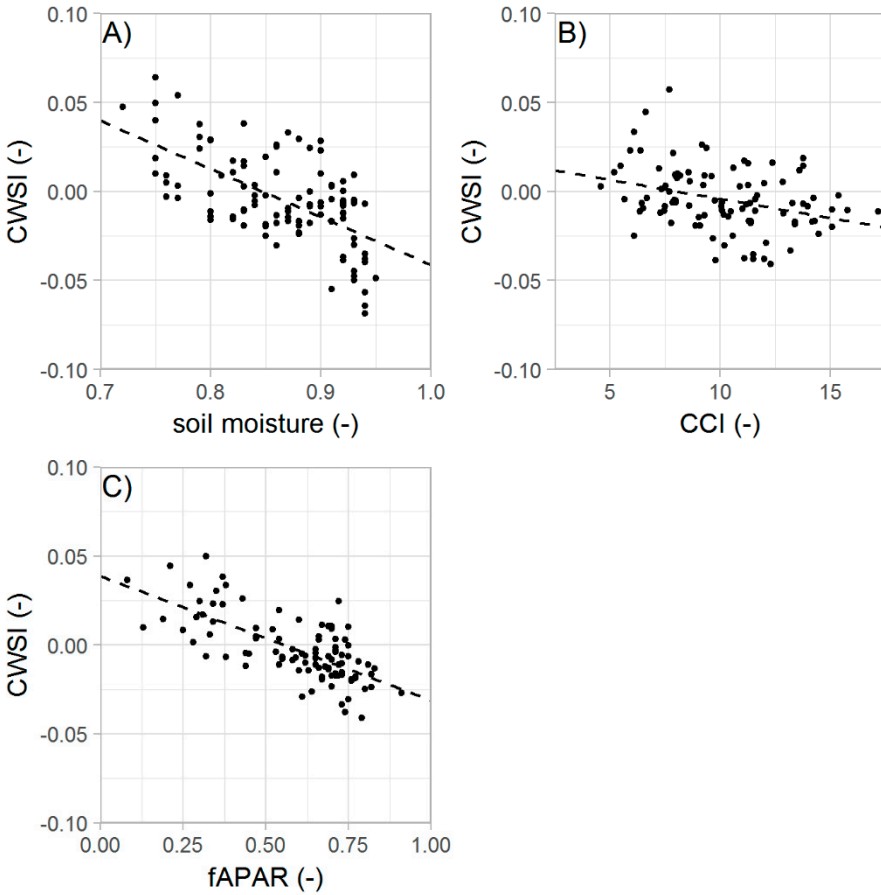

**Figure 8.** Dependence of CWSI on (**A**) soil moisture, (**B**) chlorophyll content index (CCI), and (**C**) the fraction of absorbed photosynthetically active radiation (fAPAR) for all measured plots in both research areas; the dashed line shows the linear fit.

**Table 4.** Pearson's correlation coefficient for the relation between CWSI and field measurement with its p-value and number of measurements used for calculation.

| Parameter | Number of measurements | Pearson's correlation coefficient | p-value |
|---|---|---|---|
| Soil moisture | 104 | −0.62 | <0.001 |
| CCI | 96 | −0.33 | <0.001 |
| fAPAR | 99 | −0.70 | <0.001 |

## 4. Discussion

In this section, three main aims of the study are discussed: I) Derivation of NWSB for selected wetland habitats, II) potential of using CWSI as a meteorological drought indicator for wetland habitats, and III) potential of using CWSI as a water stress indicator for these habitats.

### 4.1. The NWSB for Wetland Habitats

This study attempted to use the well-known crop water stress index in two selected wetland habitats (namely, alkaline fens, transition mires, and quaking bogs). Calculation of the CWSI requires the NWSB, which can be estimated based on a theoretical approach [41] or empirical [42] approach. Previously conducted research analyzed CWSI mainly for monitoring the health status of crops [34,41,42,47,64], fruit orchards [37,65], olive trees [39], and almond trees [40]. The authors did not find studies on CWSI on wetland habitats, especially in a temperate climate. Therefore, for the investigated habitats, the NWSB was developed (Section 3.1.). The parameters (m and b) of the NWSB for both habitats are significantly different from each other, and so are values found in the literature

for different vegetation types. This is a reflection of differences between these two habitats. Habitat 7140 in the Janów Forest Landscape Park was developed in no-runoff land lowerings and is fed by rainwater, while habitat 7230 in the Biebrza National Park is fed mainly by water from upland and the Biebrza River. In addition, different plant compositions for these habitats have an impact on the NWSB parameters. NWSB parameters (m and b) elaborated for the studied habitats differ significantly from those found in the literature for different vegetation types. These differences might be caused by different soil types (this research was conducted on organic soils) and its moisture.

The NWSB was developed based on one-day measurements in conditions described by [36]. UAS flights conducted in the scope of this research were done in similar meteorological conditions (Ta and VPD) as measurements for NWSB development. However, the variation in VPD (one of the main drivers of CWSI) was small for habitat 7230 (from 1.03 kPa to 1.87 kPA); for that reason, this obtained NWSB should be applied carefully. The field variation of VPD for habitat 7140 was also small (from 0.73 to 1.89 kPa), but measurements in the laboratory allowed the response of the habitat to be checked in a wider range of conditions (VPD from 0.38 to 2.13 kPa).

## 4.2. CWSI as Drought Indicator in Wetland Habitats

Meteorological drought is the first step of drought caused by a lack of atmospheric precipitation. Longer periods without precipitations (duration depends on habitat specification) lead to a shortage of water availability and plant stress. Comparison of the CWSI and two meteorological drought indices (Section 3.3.) shows that the CWSI is not an effective meteorological drought indicator for wetland habitats.

In the Biebrza National Park (habitat 7230), both investigated meteorological indices had similar values during UAS flights (SPI slightly above 0 in all campaigns and SCWB value ranging from –0.5 to 0) (Figure 7A); in the same time, the CWSI values between campaigns were significantly different (Figure 5A). This shows that the meteorological parameters used for SPI and SCWB calculations do not fully describe physiological drought in this habitat. Water availability is crucial for plant condition. A good indicator of water availability is groundwater level. In our research area, groundwater level changes (Figure 6A) do not correspond to meteorological drought indicators. This is due to the fact that this area is fed not only by rainwater, but also by groundwater from upland and river water.

In the Janów Forest Landscape Park (habitat 7140), meteorological indicators (SPI and SCWB) correspond better to the UAS-based CWSI (Section 3.3.). Besides one campaign, higher values of CWSI agree with lower values of SPI and SCWB. Disagreement in the third campaign is a result of the short but intensive precipitation in the day of UAS acquisition and the day after. Both indices were calculated based on the moving average. By contrast, the CWSI was calculated based on the thermal orthophotomosaic prepared for one point in time (after a long period without precipitation and/or just before intensive precipitation). For this habitat (similarly to habitat 7230), the CWSI matches the changes in groundwater level. However, due to the way of habitat water supply (mainly by precipitation), the CWSI values correspond better to drought indices than in the case of habitat 7230.

Besides the disagreement of CWSI and meteorological drought indices indications, disagreements between both meteorological indices were found. The SCWB values more often indicated drought than the SPI values. For the SPI calculation, only precipitation was used, whereas for the SCWB, evapotranspiration data were also used. The evapotranspiration is an important part of water balance in the wetland, hence the high affected indices results obtained.

## 4.3. CWSI as Water Stress Indicator in Wetland Habitats

Plant water stress is connected to a limited amount of water in the rhizosphere. Water stress is causing simultaneous changes in physiological and morphological characteristics in all plant organisms. Analysis of pigment concentrations in leaves is important in plant ecophysiological studies, providing significant information about physiological responses to environmental factors such as drought [66]. Measurements of relative chlorophyll content allow information to be obtained

on vegetation condition in a given area. On the other hand, because water deficit leads to a reduction in biomass accumulation, the criteria of plant productivity should also be used [67]. Therefore, in this research, as an indicator of water stress, groundwater level, soil moisture, fAPAR, and CCI were used. For all selected water stress indicators, their relation (strong in the case of soil moisture and fAPAR or weak but still significant for CCI) with CWSI can be observed.

Lowering of the groundwater levels in both research areas (Figure 6) correspond to the increase in observed CWSI values (Figure 5). A significant Pearson's correlation coefficient above 0.6 for the relation between the CWSI and soil moisture (Figure 8A) confirms the dependence of CWSI and water availability for plants. In wetland habitats, the optimal soil moisture is above 90%. Negative values of the CWSI for soil moisture in this range indicate no stress conditions. However, the majority of CWSI values have positive values, indicating the occurrence of water stress conditions.

It can be found in the literature that values of fAPAR and CCI correspond to changes in plant condition related to its water stress [66,67]. Its intensity depends on the stress rate and duration [56]. Fotovat et al. [59] found that by exerting severe drought stress on wheat, the chlorophyll content of a leaf significantly decreased. The chlorophyll content is one of the major factors influencing photosynthetic capacity [54,57,58], proving that it can be its reliable indicator. Droughts also affect the vegetation's capacity of intercepting solar radiation, which can be described by fAPAR. Effects of water deficit are different according to the plant's growth stage. However, a reduction in the intercepted radiation (and, therefore, in fAPAR) is always a consequence of drought [54,55].

Therefore, the correlation of measured biophysical parameters and CWSI (Figure 8B,C) supports the hypothesis that the CWSI can be used in wetland habitats as a water stress indicator. However, in this research, weak CWSI correlation with CCI was due to the fact that the measurements were performed not for each plant or species present in the research plot but for the dominant species only. CCI values differ not only between species but also between individuals. Despite the measurements performed for each research plot, 10 individuals for each dominant in triplicate (on three different leaves) obtained CCI values (even averaged) do not correspond accurately to the averaged CWSI values obtained from UAS data, as they do not represent the whole variability occurring in a given plot. The reason is also the inability to perform CCI measurements for bryophytes, whose percentage in the analyzed area was quite large (80%–100%). However, CCI measurements performed in this study on dominant plant species and other field observations were used to monitor condition changes in time, allowing us to also expect changes in CWSI values.

## 5. Conclusions

In the scope of this research, new NWSBs for two non-forest Natura 2000 habitats (code 7140 and 7230) were developed. Obtained NWSB parameters for both habitats were different. This is a reflection of the high diversity of wetland habitats. The further application of CWSI should be supported with ground truth data collection and, in the case of different wetland habitats types, preceded by the NWSB derivation.

The results of the analysis clearly show that CWSI cannot be used as a meteorological drought indicator. It results in the fact that these indices are calculated based on precipitation and evapotranspiration. The wetland areas are fed not only by rainwater; hence, the lack of precipitation does not have to lead to water stress. Moreover, evapotranspiration plays an important role in the wetlands, but its direct measurements are still challenging and all estimations are biased. Hence, simple metrological drought indicators might describe wetland conditions incorrectly.

Biophysical parameters describing plant condition related to its water stress (fAPAR and CCI) significantly correlated with the CWSI values. The correlation of CCI was not as strong as for fAPAR, but it was a result of the used measurement methodology. Further analyses should include bryophytes for habitats that are characterized by their high coverage. However, due to their morphology, this requires specialized sensors, and measurement in the field becomes even more time-consuming. Together with CWSI correlation with soil moisture and the correspondence of CWSI values to fluctuations in the groundwater level, it shows that the CWSI can be used in the wetland habitat as a water stress indicator.

**Author Contributions:** Conceptualization, Jarosław Chormański; Data curation, Wojciech Ciężkowski, Sylwia Szporak-Wasilewska and Jacek Jóźwiak; Formal analysis, Wojciech Ciężkowski; Funding acquisition, Jarosław Chormański; Investigation, Wojciech Ciężkowski, Sylwia Szporak-Wasilewska, Małgorzata Kleniewska, Jacek Jóźwiak, Tomasz Gnatowski, Piotr Dąbrowski, Maciej Góraj, Jan Szatyłowicz and Jarosław Chormański; Methodology, Wojciech Ciężkowski, Sylwia Szporak-Wasilewska, Małgorzata Kleniewska, Jacek Jóźwiak, Tomasz Gnatowski, Piotr Dąbrowski, Maciej Góraj, Jan Szatyłowicz and Jarosław Chormański; Project administration, Stefan Ignar; Resources, Wojciech Ciężkowski and Jacek Jóźwiak; Software, Wojciech Ciężkowski; Supervision, Stefan Ignar and Jarosław Chormański; Validation, Wojciech Ciężkowski; Visualization, Wojciech Ciężkowski; Writing – original draft, Wojciech Ciężkowski; Writing – review & editing, Sylwia Szporak-Wasilewska, Małgorzata Kleniewska, Jacek Jóźwiak, Tomasz Gnatowski, Piotr Dąbrowski, Maciej Góraj, Jan Szatyłowicz and Jarosław Chormański. All authors have read and agreed to the published version of the manuscript.

**Funding:** The study was co-financed by the Polish National Centre for Research and Development (NCBiR) and MGGP Aero under the program "Natural Environment, Agriculture and Forestry" BIOSTRATEG II.: The innovative approach supporting monitoring of non-forest Natura 2000 habitats, using remote sensing methods (HabitARS), project number: DZP/BIOSTRATEG-II/390/2015. The Consortium Leader is MGGP Aero. The project partners include the University of Lodz, the University of Warsaw, Warsaw University of Life Sciences, the Institute of Technology and Life Sciences, the University of Silesia in Katowice, Warsaw University of Technology.

**Acknowledgments:** Meteorological data were made available by the Institute of Meteorology and Water Management, National Research Institute (IMGW-PIB). We thank Jonathan C-W Chan for comments that greatly improved the manuscript. We thank anonymous reviewers whose comments/suggestions helped improve and clarify this manuscript. Publication of the article has been financed by the Polish National Agency for Academic Exchange under the Foreign Promotion Programme.

**Conflicts of Interest:** The authors declare no conflict of interest.

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
