# Peer review of "Remotely Sensed Land Surface Temperature-Based Water Stress Index for Wetland Habitats"

_remotesensing, doi:10.3390/rs12040631_

Round 1
Reviewer 1 Report
The authors have addressed my previous concerns in a decent way.
Author Response
The authors have addressed my previous concerns in a decent way.
The authors kindly thanks you for the positive review.
Reviewer 2 Report
This study applied CWSI to two wetland habitats to detect drought and water stress. Two meteorology drought indices were also calculated to validate the CWSI detected drought. This study found that CWSI is not an effective indicator of meteorology drought, but good at water stress detect.
Generally, the manuscript is well written and easy to follow, but there are some key weakness should be addressed before it can be accepted.
The measurement of LST. In CWSI, Tc is the canopy temperature, but the measurement from UAV is the surface temperature which includes both canopy temperature and the background temperature (water temperature in this case?). I think this is the key issue I have with this study. The measured LST should be further processed to seperate canopy temperature and background temperature. Otherwise, the noise from background will strongly impact the analysis and the conclusion from this study. The upper limit of dT. It is not clear how the upper limit of dT was calculated and the results were not shown and discussed in the study. However, this parameter is important that can affect the value of CWSI. This should be clearly demonstrate in the study. SPI and SCWB. They were discussed later in the manuscript, but not in the method part. Method of how to estimate SPI and SCWB should be included in the method section. More information about why choose fAPAR, CCI and soil moisture as indicator of water stress should be included. Are they really enough to demonstrate vegetation water stress? A lot of studies used leaf water content. Whether the three indicators show similar stress as leaf water content? Suggest adding a section of discussion about the difference between using CWSI to wetland and to agriculture. Since CWSI was initially created for crop area, whether it is applicable to wetland? Some miner edits: Some typos need to be corrected, for example, line 167, E should be e. More information about CWSI should be added in the introduction.Author Response
This study applied CWSI to two wetland habitats to detect drought and water stress. Two meteorology drought indices were also calculated to validate the CWSI detected drought. This study found that CWSI is not an effective indicator of meteorology drought, but good at water stress detect.
Generally, the manuscript is well written and easy to follow, but there are some key weakness should be addressed before it can be accepted.
Authors would like to thank for useful comments which allowed to prepare a new improved version of the manuscript. Below reviewer can find detailed answers to the comments.
The measurement of LST. In CWSI, Tc is the canopy temperature, but the measurement from UAV is the surface temperature which includes both canopy temperature and the background temperature (water temperature in this case?). I think this is the key issue I have with this study. The measured LST should be further processed to seperate canopy temperature and background temperature. Otherwise, the noise from background will strongly impact the analysis and the conclusion from this study.
The wetland habitats analyzed in this study have dense plant coverage, typically plants cover 80-100% of the background which in turn is mainly represented by bryophytes or necromass (dead organic matter). The research using UAS was carried out outside the period of flooding occurrence, with full vegetation development in the period from 24.07 to 18.09 in the Biebrza river valley and from 14.07 to 09.09 in the Janowskie Forest. Hence, in this study we assumed that LST is equal to canopy temperature which is robust assumption in authors opinion. We added an explanation about using LST as canopy temperature at the beginning of section 2.3.
The upper limit of dT. It is not clear how the upper limit of dT was calculated and the results were not shown and discussed in the study. However, this parameter is important that can affect the value of CWSI. This should be clearly demonstrate in the study.
Authors adapted the methodology describes by Taghvaeian et al. 2012 [36]. To clarify dT upper limits calculation, authors added equation 4 in section 2.2.1. and extend description.
Taghvaeian, S.; Chávez, J. L.; Hansen, N. C. Infrared thermometry to estimate crop water stress index and water use of irrigated maize in Northeastern Colorado. Remote Sens. 2012, 4, 3619–3637.
SPI and SCWB. They were discussed later in the manuscript, but not in the method part. Method of how to estimate SPI and SCWB should be included in the method section.
The obtained results of SPI and SCWB indices were discussed in the Results section (3.3.) and described in the Materials and Methods (section 2.4. Meteorological drought indices).
More information about why choose fAPAR, CCI and soil moisture as indicator of water stress should be included. Are they really enough to demonstrate vegetation water stress? A lot of studies used leaf water content. Whether the three indicators show similar stress as leaf water content?
In our opinion, these 3 parameters are fully sufficient to determine the overdrying of the given wetland habitats. Please note that these three parameters describe the state of soil moisture, the reaction of individual plants (CCI) and the entire patch of the community (fAPAR).
The reviewer rightly noted that leaf water content is one of the best indicators of changes in plant moisture because it directly indicates the water content in tissues. Another equally precise method is the determination of the leaf suction pressure. However, these methods also have their drawbacks. First of all, they are destructive. So their use in protected areas or measuring protected species is formally difficult. They are also very laborious and time-consuming methods. If this is not a problem with pot or greenhouse experiments, it is very difficult for studies carried out on wetlands. Of course in the MM section (2.5. Biophysical parameters, soil moisture and groundwater level), we added more information about the selection of fAPAR and CCI.
Suggest adding a section of discussion about the difference between using CWSI to wetland and to agriculture. Since CWSI was initially created for crop area, whether it is applicable to wetland?
In our paper, we discuss the potential of using CWSI on wetlands. Application of this index (regardless of the area of application) requires the NWSB developed for studied habitat.
The applicability is also confirmed by the received dependencies between CWSI and soil moisture and fAPAR, and week but significant correlation with CCI.
Some minor edits: Some typos need to be corrected, for example, line 167, E should be e. More information about CWSI should be added in the introduction.
The manuscript was rewritten, in the new version we have tried our best to correct all indicated errors.
Reviewer 3 Report
This paper tries to describe the adaption of the thermal water stress index also known as Crop Water Stress Index (CWSI) for wetlands. The work is based on the lab experiments and field measurements. The methodology is proper and comprehensive. However, the section 3.3 does not present the correct results and the reviewer believe the results do not strongly support the conclusion that CWSI is a good index of water stress/drought. Therefore, the reviewer does not suggest publication and encourage to consider re-submitting it with the improvements.
Introduction:
Line 71: the paper refers here can be used to evaluate the paper’s contribution
Line 73-75: The sentence is not grammatically corrected, should be corrected to make it more comprehensive.
Section 3.3:
Line 340: the groundwater level was highest (10cm), the reviewer considers it to be -10cm more appropriate.
Line 366-368: The numbers “-0.68, -0.57 and -0.61” for first, second and third campaign do not match the Figure 7; the same mismatch happens for SCBW. Comparing with Table1, the date 18.9.2016 seems not locate in the third vertical line in the Figure 7A.
Line 372-375: The same problem as the previous paragraph, the reviewer cannot find the SPI/SCWB values in the Figure 7B as described in the paragraph. The author should carefully make sure the images used are correct.
Section 3.4:
As the title of this paper describes, this section is expected to be the core section to prove the effectiveness of CWSI as the water stress indicator for the wetland ecosystems. However, this section needs to be more comprehensive. The correlation coefficient between CWSI and CCI is not too strong.
Section 4.1:
The author mentioned the parameters (m and b) for both habitats are significantly different due to vegetation types or the soil types and its moisture. The further discussion should be included with such difference’s impact on the CWSI performance as a water stress index.
The Citation [29] “The Challenges of Remote Monitoring of Wetlands” ( https://www.mdpi.com/2072-4292/7/8/10938/htm) stated the problems of wetland remote sensing:
One main factor is that wetlands are not unified by a common land-cover type or vegetation form in the way that forests are populated with trees, grasslands with grasses, and shrub-lands with shrubs. A second factor that makes wetlands so challenging to map remotely is that they are highly dynamic in ways that substantially alter their reflectances and energy back-scatter properties, sometimes within hours or days. A third factor is that steep environmental gradients in and around the edges of wetlands produce narrow ecotones that are often below the spatial resolving capacity of remote sensors.The manuscript should discuss the contribution of this work to help solve listed problems specially for the wetlands.
Sections 4.2 and 4.3:
The sections 4.2 and 4.3 are based on the sections 3.3 and 3.4, where the concerns should be solved to make sections 4.2 and 4.3 more plausible.
The conclusion section should be formatted with same indent and be strengthened. The presented results do not strongly support the conclusion.
Author Response
Authors would like to thank for useful comments which allows preparing a new version of the manuscript. Below you can find detailed answers to the comments.
This paper tries to describe the adaption of the thermal water stress index also known as Crop Water Stress Index (CWSI) for wetlands. The work is based on the lab experiments and field measurements. The methodology is proper and comprehensive. However, the section 3.3 does not present the correct results and the reviewer believe the results do not strongly support the conclusion that CWSI is a good index of water stress/drought. Therefore, the reviewer does not suggest publication and encourage to consider re-submitting it with the improvements.
Authors carefully rewrote section: 3.3. Meteorological parameters and drought indices. Discussion and conclusions regarding the results were also corrected. We have clearly indicated that the CWSI method cannot be used as demonstrated by our analysis as a meteorological drought indicator. In the discussion, we provided more details about this.
Results in section: 3.4. Correlation of CWSI and field measurements, in authors opinions CWSI can be used as a water stress indicator in wetlands. The strong correlation of CWSI with biophysical parameters that represent habitat and/or plant condition (soil moisture and fAPAR), and week but significant correlation with CCI prove this. However, the low correlation with CCI is caused by limitations of the measurement method used and was described in more detail in the discussion.
Introduction:
Line 71: the paper refers here can be used to evaluate the paper’s contribution
Line 73-75: The sentence is not grammatically corrected, should be corrected to make it more comprehensive.
We tried to do our best to correct all grammatical errors.
Section 3.3:
Line 340: the groundwater level was highest (10cm), the reviewer considers it to be -10cm more appropriate.
We have changed values of groundwater level according to reviewer’s suggestion.
Line 366-368: The numbers “-0.68, -0.57 and -0.61” for first, second and third campaign do not match the Figure 7; the same mismatch happens for SCBW. Comparing with Table1, the date 18.9.2016 seems not locate in the third vertical line in the Figure 7A.
Line 372-375: The same problem as the previous paragraph, the reviewer cannot find the SPI/SCWB values in the Figure 7B as described in the paragraph. The author should carefully make sure the images used are correct.
Authors rewrote sections about meteorological drought indices and created new figures. Now the information in text, tables and figures corresponds to each other.
Section 3.4:
As the title of this paper describes, this section is expected to be the core section to prove the effectiveness of CWSI as the water stress indicator for the wetland ecosystems. However, this section needs to be more comprehensive. The correlation coefficient between CWSI and CCI is not too strong.
Two of three used parameters (fAPAR and soil moisture) strongly correlated with CWSI. In our opinion, due this fact we can state that CWSI can be used in wetlands as a water stress indicator. Much lower correlation of CWSI and CCI is a result of the used measurement technique which is described in more detail in the discussion (section 4.3).
Section 4.1:
The author mentioned the parameters (m and b) for both habitats are significantly different due to vegetation types or the soil types and its moisture. The further discussion should be included with such difference’s impact on the CWSI performance as a water stress index.
Authors extended the discussion about NWSB for wetland habitats (section 4.1 ) according to reviewer’s suggestion.
The Citation [29] “The Challenges of Remote Monitoring of Wetlands” ( https://www.mdpi.com/2072-4292/7/8/10938/htm) stated the problems of wetland remote sensing:
One main factor is that wetlands are not unified by a common land-cover type or vegetation form in the way that forests are populated with trees, grasslands with grasses, and shrub-lands with shrubs. A second factor that makes wetlands so challenging to map remotely is that they are highly dynamic in ways that substantially alter their reflectances and energy back-scatter properties, sometimes within hours or days. A third factor is that steep environmental gradients in and around the edges of wetlands produce narrow ecotones that are often below the spatial resolving capacity of remote sensors.
The manuscript should discuss the contribution of this work to help solve listed problems specially for the wetlands.
In the scope of the article, authors undertook the task of applying CWSI to selected wetland habitats. This research is an initial step for further analyses of water stress in wetlands. However, problems mentioned in the citation [29] require more comprehensive analyses for different wetland types which in itself can be the next step in the team’s research.
Sections 4.2 and 4.3:
The sections 4.2 and 4.3 are based on the sections 3.3 and 3.4, where the concerns should be solved to make sections 4.2 and 4.3 more plausible.
All sections were corrected according to reviewer’s comments.
The conclusion section should be formatted with same indent and be strengthened. The presented results do not strongly support the conclusion.
Authors took into account most of the comments and in the newest version of the manuscript conclusions were corrected to correspond with all presented results.
Round 2
Reviewer 2 Report
The authors addressed most of my concerns and I think it is good to be published.
Reviewer 3 Report
I acknowledge the authors effort in addressing all my comments. The paper is now acceptable to me in its current form.